# POSITION-AWARE SINGULAR VALUE SHRINKAGE FOR UNFOLDED DYNAMIC MRI RECONSTRUCTION

## ABSTRACT

Dynamic MRI reconstruction benefits from low-rank priors to exploit spatiotemporal redundancy. Recent deep unfolding networks (DUNs) often adopt Singular Value Thresholding (SVT) to apply low-rank constraints. However, most methods apply uniform or globally scaled thresholds, ignoring the unequal importance of singular values and the resolution-dependent nature of dynamic MR images. This leads to suboptimal shrinkage and poor generalization across anatomical variations. Existing adaptive shrinkage techniques in classical models are not trainable and incompatible with end-to-end learning. To address these challenges, we propose a Position-Aware Adaptive Singular-Value Shrinkage (PASS) module that learns to perform context-aware SVT using spectral positional encoding and a neural gating mechanism. This enables selective preservation of important components while suppressing noise and redundancy. We integrate PASS into a deep unfolding network based on low-rank plus sparse decomposition, and introduce a multi-resolution training strategy to improve the adaptivity of PASS across varying anatomical scales and acquisition settings. Experimental results on two dynamic cardiac MRI datasets demonstrate that our method achieves superior reconstruction quality and generalization compared to existing SVT-based baselines. **Our code will be available after acceptance.**

## 1 INTRODUCTION

Dynamic magnetic resonance imaging (MRI) captures temporal changes in anatomical structures and plays a critical role in clinical tasks such as perfusion analysis Geon-Ho et al. (2014), cardiac imaging Chen et al. (2020); Wang et al. (2021a), and medical report generation Wang et al. (2024). To reconstruct high-quality dynamic image sequences, a large amount of k-space data must be acquired. However, conventional MRI requires inherently long acquisition times, increasing patient discomfort and exacerbating motion-induced artifacts. To accelerate the scan, contemporary techniques reconstruct full-resolution images from undersampled k-space data. Among these methods, algorithms that embed structural priors such as low-rank and sparsity constraints have gained prominence, as they exploit the pronounced spatiotemporal redundancy and global correlations characteristic of dynamic MRI sequences.

Singular Value Thresholding (SVT) is effective in suppressing noise and promoting low-rank structure by shrinking the singular values of feature matrices or tensors. Due to its simplicity and effectiveness, SVT is widely used in low-rank regularized models. In particular, it is commonly integrated into deep unfolding networks (DUNs) with low-rank priors Ke et al. (2021); Zhang et al. (2024; 2025), and the formulation used in them is:

$$\tau = \text{Sigmoid}(r) \times \sigma_1, \qquad \sigma_i = \text{Max}(\sigma_i - \tau, 0), \tag{1}$$

where $r$ is a learnable scalar, $\sigma_i$ is $i$-th largest singular value.

However, existing SVT-based methods face three key limitations. **1) Uniform shrinkage without discrimination:** Most deep unfolding networks incorporate SVT with a global threshold, often parameterized as a scaled version of the largest singular value. This approach assumes that all singular values contribute equally to the underlying image structure. In practice, large singular values often capture dominant anatomical information, while small ones are more likely to represent noise or subtle textures. Applying a uniform threshold disregards this spectral hierarchy and leads to either

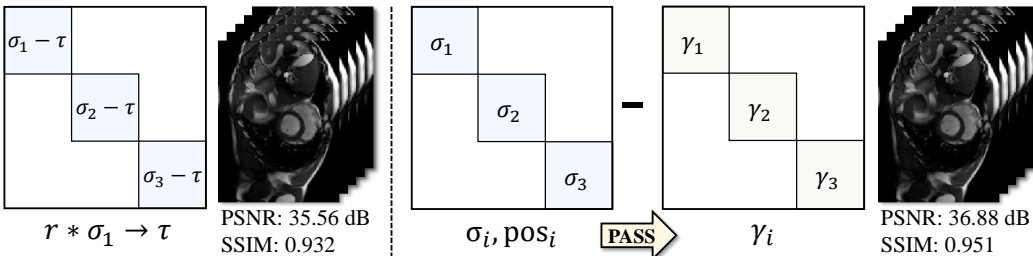

Figure 1: Comparison between uniform threshold (left) and individual threshold generated by the proposed PASS (right). Here, $\sigma_i$ denotes the $i$-th largest singular value, $\text{pos}_i$ is its position encoding, and $\gamma_i$ is the individual threshold.

excessive suppression of fine details or insufficient attenuation of noise. **2) Lack of resolution adaptivity:** In dynamic MRI, image resolution varies across datasets due to scanner configurations, field-of-view settings, and anatomical coverage. These variations lead to significantly different singular value distributions. Existing methods apply the same shrinkage rule across all inputs, regardless of their spectral profiles. This rigid strategy often fails to regularize appropriately: high-resolution inputs may retain unwanted high-frequency noise, while low-resolution inputs may lose important structural content. The inability to adapt across resolution scales limits model robustness in practical applications. **3) Incompatibility of traditional adaptive methods with deep learning:** Methods like the Adaptive Trace Norm (ATN) and NNFN regularizer introduce handcrafted adaptive shrinkage rules based on SURE estimation and norm discrepancy respectively, to avoid per-value tuning Gavish & Donoho (2017); Wang et al. (2021b). Although these techniques show effectiveness in model-based reconstruction, they rely on hand-crafted formulations and are designed for fixed-size matrices. They do not generalize well to variable-size inputs common in dynamic MRI. Furthermore, they lack trainable parameters and cannot be integrated into end-to-end deep networks. Their inflexibility prevents them from leveraging large-scale data and learning spatially adaptive priors.

To address these challenges, we propose a Position-Aware Adaptive Singular-Value Shrinkage (PASS) mechanism. PASS enhances the SVT operator by integrating positional encoding and a learnable gating module to enable context-sensitive shrinkage. Each singular value is assigned a continuous position score reflecting its relative importance. These scores and magnitudes are jointly processed to generate individualized shrinkage factors (Fig. 1). Important components are preserved, while noisy or less informative ones are suppressed. To support adaptivity, we introduce a multi-resolution training strategy that randomly crops inputs at varying resolutions. This exposes the model to diverse structural scales and spectral distributions, helping PASS adjust shrinkage based on spatial context and spectral semantics.

We integrate PASS into a deep unfolding network based on the Low-Rank plus Sparse (L+S) decomposition model Huang et al. (2021). The network alternates between low-rank and sparse estimation, both guided by learnable priors. To enhance their interplay, we introduce an adaptive regularization strategy that adjusts the relative weights of low-rank and sparse terms based on local features such as gradients and textures. Low-rank regularization dominates in structured regions like brain and heart, while sparse regularization is emphasized in fine-detail areas like vessels and lesions. Our key contributions are as follows:

- We propose a novel PASS mechanism that introduces position encoding and learnable gating to improve the discriminative capacity of singular value thresholding. To support its resolution-awareness, we further adopt a multi-resolution training strategy that exposes the PASS to inputs with varying spatial resolutions.

- We integrate PASS into a deep unfolding network based on the L+S decomposition, enhancing the interaction between low-rank and sparse components through adaptive regularization. The model dynamically adjusts the strength of each prior according to local structural features, improving reconstruction fidelity across diverse anatomical regions.

- We conduct extensive experiments on dynamic MRI datasets, demonstrating that our method achieves superior reconstruction performance over existing low-rank and adaptive thresholding baselines.

## 2 RELATED WORK

### 2.1 GENERAL RECONSTRUCTION MODEL OF DYNAMIC MRI

In k-t undersampling for Cartesian 3D dynamic MRI images, the reconstruction process can be formulated as:

$$\mathbf{d} = \mathscr{A}\mathcal{X} + \mathbf{n}, \tag{2}$$

where $\mathcal{X} \in \mathbb{C}^{T \times H \times W}$ represents the dynamic MRI data to be reconstructed, the variable $\mathbf{d} \in \mathbb{C}^{m}$ corresponds to the undersampled $k$-space data, and $\mathbf{n} \in \mathbb{C}^{m}$ is Gaussian noise. $H$ and $W$ represent the spatial dimensions, and $T$ represents the temporal dimension. The operator $\mathscr{A} : \mathbb{C}^{T \times H \times W} \to \mathbb{C}^{m}$ is the encoding operator, which has different forms depending on the specific application. In this context, $\mathscr{A} = \mathscr{U}\mathscr{F}\mathscr{C}$, where $\mathscr{U} : \mathbb{C}^{T \times H \times W} \to \mathbb{C}^{m}$ is the sampling operator, $\mathscr{F} : \mathbb{C}^{T \times H \times W} \to \mathbb{C}^{T \times H \times W}$ represents the 2D spatial Fourier transform applied along the $H$ and $W$ axes, $\mathscr{C} : \mathbb{C}^{T \times H \times W} \to \mathbb{C}^{T \times H \times W \times C}$ is the coil sensitivity maps for multi-coil MRI with $C$ coils. In the case of a single coil, $\mathscr{C}$ simplifies to the identity transform.

The problem of reconstructing dynamic MRI is typically formulated as an optimization problem:

$$\min_{\mathcal{X}} \frac{1}{2}\|\mathscr{A}\mathcal{X} - \mathbf{d}\|_2^2 + \mathrm{R}(\mathcal{X}), \tag{3}$$

where $\mathrm{R}(\mathcal{X})$ is the regularization term, and $\|\mathscr{A}\mathcal{X} - \mathbf{d}\|_2^2$ is the data fidelity term. When the model incorporates the sparsity of the data, the regularization term is often written as $\|\mathcal{X}\|_1$. The tensor $\ell_1$ norm is defined as the sum of absolute values of all elements in the tensor. Some methods also leverage the low-rank property for reconstruction, in which case the regularization term is typically expressed as $\|\mathcal{X}\|_*$, where $\|\cdot\|_*$ denotes the tensor nuclear norm.

### 2.2 SINGULAR VALUE SHRINKAGE AND THRESHOLDING

Singular value shrinkage and thresholding (SVT) methods have been widely adopted for low-rank matrix recovery, with applications in various fields including image denoising and matrix completion. The classical hard thresholding approach truncates singular values below a fixed threshold Gavish & Donoho (2014), effectively discarding noise but also potentially removing important components. Soft thresholding applies a continuous shrinkage function allowing singular values to be reduced smoothly Cai et al. (2010), but it treats all singular values equally, which limits their flexibility in handling the varying importance of singular values in different regions of the data.

Adaptive shrinkage modifies singular values according to their individual magnitudes, allowing for more tailored shrinkage. For example, the ATN method proposed by Josse and Sardy Gavish & Donoho (2017) introduces a flexible family of shrinkage functions with parameters automatically tuned via SURE, showing strong robustness across varying noise levels and matrix ranks. Similarly, Wang et al. Wang et al. (2021b); Li et al. (2025) propose the NNFN regularizer, which achieves adaptive shrinkage by penalizing the difference between the nuclear norm and Frobenius norm. This technique can improve performance by selectively shrinking less important singular values while preserving significant ones. However, most existing adaptive shrinkage methods still rely on manually set thresholds or handcrafted penalties and often ignore the spatial and semantic context of the data. These methods are rarely integrated into deep learning frameworks, where learnable priors could further enhance flexibility and data adaptability. Furthermore, these methods are not directly compatible with deep learning pipelines, especially in dynamic MRI reconstruction, where input sizes can vary across patients.

To address these issues, we propose a PASS mechanism. PASS enhances traditional SVT by incorporating position encoding and learnable gating mechanisms, allowing for adaptive shrinkage of singular values based on their rank and local image context. This enables selective suppression of less important singular values while preserving critical image structures. By embedding PASS within a deep unfolding network, we achieve an end-to-end trainable framework that dynamically adjusts low-rank and sparse priors based on the local image features such as textures and edges.

## 3 METHODOLOGY

### 3.1 MODEL DERIVATION

The $\mathcal{L} + \mathcal{S}$ model Chandrasekaran et al. (2011) decomposes the tensor $\mathcal{X}$ into two components: a dynamic component $\mathcal{S}$ and a background component $\mathcal{L}$. The dynamic part $\mathcal{S}$ captures the fine details of the image and is typically sparser than $\mathcal{X}$, while the background component $\mathcal{L}$ exhibits strong temporal correlations, often resulting in a lower rank. Based on this decomposition, the reconstruction model from equation 3 can be formulated as:

$$\min_{\mathcal{X}} \frac{1}{2}\|\mathscr{A}\mathcal{X} - \mathbf{d}\|_2^2 + \lambda_1\|\mathcal{L}\|_{\Phi*} + \lambda_2\|\mathcal{S}\|_{\Psi 1},$$
$$\text{s.t. } \mathcal{X} = \mathcal{W} \odot \mathcal{L} + (1 - \mathcal{W}) \odot \mathcal{S}, \tag{4}$$

where $\mathcal{X} \in \mathbb{C}^{T \times H \times W}$, $\lambda_1$ and $\lambda_2$ are regularization parameters, and $\mathcal{W} \in [0, 1]^{T \times H \times W}$ is a learnable weight tensor that adaptively balances the contributions of $\mathcal{L}$ and $\mathcal{S}$ across spatial-temporal locations. The operator $\odot$ denotes element-wise multiplication.

The low-rank term $\|\mathcal{L}\|_{\Phi*}$ is defined as a transformed tensor nuclear norm (TTNN) Zhang et al. (2024), under the t-SVD framework Kilmer & Martin (2011) that encourages low-rank structure. It is formulated as:

$$\|\mathcal{L}\|_{\Phi*} = \sum_{i=1}^{T} \|\Phi(\mathcal{L})^{(i)}\|_*.$$

The transformation $\Phi$ is a convolutional neural network (CNN) that maps the input tensor into a latent space where low-rankness is more evident. For a transformed tensor $\mathcal{Y} = \Phi(\mathcal{L})$, the notation $\mathcal{Y}^{(i)}$ refers to its $i$-th frontal slice, i.e., $\mathcal{Y}^{(i)} = \mathcal{Y}(:, :, i)$.

Similarly, $\|\cdot\|_{\Psi 1}$ represents a transformed tensor $\ell_1$-norm that promotes sparsity. It formulated as:

$$\|\mathcal{S}\|_{\Psi 1} = \|\Psi(\mathcal{S})\|_1.$$

The mapping $\Psi$ is also implemented as a CNN and projects the input tensor into a sparse representation domain. These learnable transformations $\Phi$ and $\Psi$ enable adaptive regularization by aligning with the intrinsic low-rank and sparse properties of the data.

To solve the optimization problem equation 4, we rewrite it using Lagrangian multipliers as follows:

$$L(X, \mathcal{L}, \mathcal{S}, \mathcal{Y}) = \frac{1}{2}\|\mathscr{A}\mathcal{X} - \mathbf{d}\|_F^2 + \lambda_1\|\mathcal{L}\|_{\Phi*} + \lambda_2\|\mathcal{S}\|_{\Psi 1}$$
$$+ <\mathcal{Y}, \mathcal{X} - \mathcal{T}> + \frac{\mu}{2}\|\mathcal{X} - \mathcal{T}\|_F^2, \tag{5}$$

where $\mathcal{T} = \mathcal{W} \odot \mathcal{L} + (1 - \mathcal{W}) \odot \mathcal{S}$, $\mathcal{Y}$ is Lagrangian multiplier, and $\mu > 0$ is penalty parameter.

Unlike conventional L+S decomposition, this weighted formulation enhances the interaction between the low-rank and sparse components by avoiding simple linear fusion. Instead, it enables adaptive regularization that dynamically adjusts their relative contributions based on the spatial and temporal context. Specifically, low-rank regularization dominates in regions with coherent structures, while sparse regularization intensifies in areas with sharp transitions.

To efficiently solve equation 5, we adopt a composite splitting algorithm Huang et al. (2011); Han (2003), which decomposes the problem into simpler subproblems. The formulas of each subproblem are as follows:

$$\begin{cases} \tilde{\mathcal{X}} = \mathcal{X} - \mu_1 \mathscr{A}^H(\mathscr{A}(\mathcal{X}) - \mathbf{d}), & \mathcal{L} = \arg\min_{\mathcal{L}} \frac{\mathcal{W}}{2\mu_2}\|\mathcal{X} - \tilde{\mathcal{X}}\|_F^2 + \lambda_1\|\mathcal{L}\|_{\Phi*}, \\ \mathcal{S} = \arg\min_{\mathcal{S}} \frac{1-\mathcal{W}}{2\mu_3}\|\mathcal{X} - \tilde{\mathcal{X}}\|_F^2 + \lambda_2\|\mathcal{S}\|_{\Psi 1}, & \mathcal{X} = \mathcal{W} \odot \mathcal{L} + (1 - \mathcal{W}) \odot \mathcal{S}. \end{cases} \tag{6}$$

For simplicity, we omit the superscript $k$ that indicates the $k$-th iteration update, both here and the subsequent formulas.

Next, we unfold the iterative steps of equation 6 into a deep neural network as Fig. 2, enabling end-to-end optimization and data-driven regularization. This unfolding not only accelerates inference but also embeds the optimization prior into a trainable architecture. In the following, we detail how each subproblem in the original solver is implemented as a corresponding module in the network.

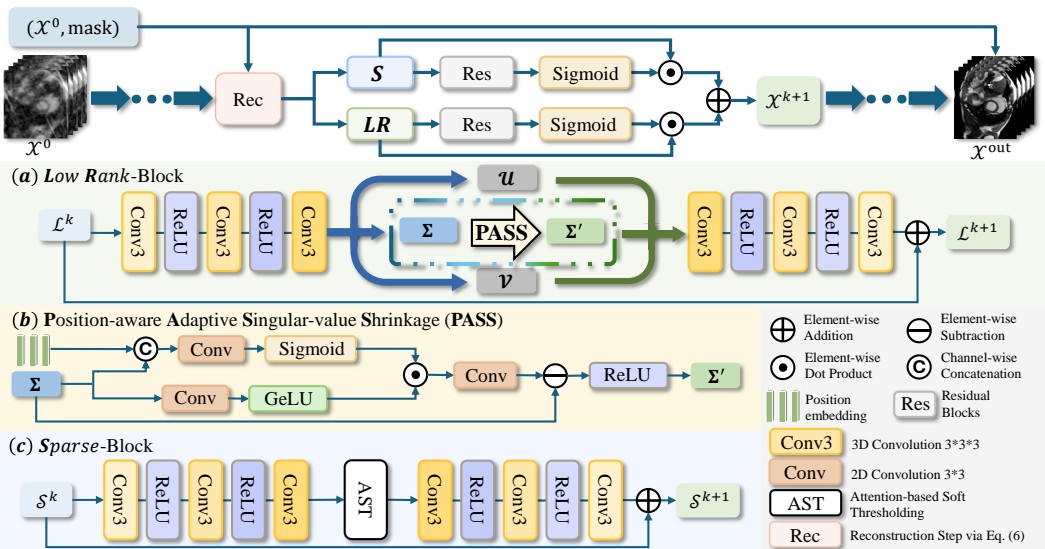

Figure 2: PASS network framework. (a) Low Rank Block, (b) PASS Block, (c) Sparse Block.

### 3.1.1 THE $\mathcal{L}$ LAYER WITH POSITION-AWARE ADAPTIVE SHRINKAGE

For the $\mathcal{L}$ subproblem in the unfolded network, many methods use the standard SVT operator. Given a tensor input, the tensor nuclear norm is computed via SVD on each frontal slice, followed by uniform shrinkage of singular values to suppress noise. A typical formulation is shown in equation 1. This assumes all singular values contribute equally and applies a fixed threshold across them. However, uniform shrinkage fails to distinguish informative components from noise. In dynamic MRI, large singular values often capture key anatomical structures, while smaller ones may reflect noise or weak textures. Uniform suppression may remove useful details or retain noisy elements, reducing regularization effectiveness in structurally diverse regions.

To overcome these limitations, we propose a Position-Aware Adaptive Singular-value Shrinkage (PASS) mechanism. The core idea is to enhance the SVT operator by introducing semantic and positional awareness. Instead of applying a fixed threshold across all singular values, PASS generates an adaptive shrinkage profile for each singular value using a data-driven gating function.

Given the transformed tensor slice $\mathcal{F}_1^{(i)} = \Phi(\mathcal{X})^{(i)}$, we first perform SVD as follows:

$$\mathcal{F}_1^{(i)} = U\Sigma V^H, \quad \Sigma = \mathrm{diag}(\sigma_1, \ldots, \sigma_K), \tag{7}$$

where singular values $\sigma_j$ are sorted in descending order. PASS adaptively computes a shrinkage value $\gamma_j$ for each singular value, modifying it according to:

$$\sigma'_j = \max(\sigma_j - \gamma_j, 0), \quad j = 1, 2, \ldots, K. \tag{8}$$

To achieve adaptive shrinkage, we leverage two complementary sources of information: singular value magnitudes and their spectral positions. First, positional embeddings $p_j \in [0, 1]$ encode the relative spectral positions of singular values using a sinusoidal formulation as follows:

$$p_j = \sin\left(\frac{\mathrm{linspace}(1, 0, K)}{10000^{j/K}}\right). \tag{9}$$

These embeddings form a positional tensor $P = [p_1, p_2, \ldots, p_K]$ shared across batches and temporal dimensions. These positional embeddings capture the hypothesis that different singular values $\sigma = [\sigma_1, \sigma_2, \ldots, \sigma_K]$ have distinct semantic importance according to their relative positions. We then concatenate the singular values $\Sigma$ and their corresponding positional embeddings $P$ to form a combined tensor $\Lambda \in \mathbb{R}^{B \times T \times K \times 2}$, where $B$ and $T$ denote the batch and temporal dimensions respectively, as follows:

$$\Lambda = [\sigma, P]. \tag{10}$$

We process this concatenated tensor through several convolutional layers and a sigmoid activation, producing a shrinkage ratio vector $r_j$ as follows:

$$r_j = \text{Sigmoid}(\text{Convs}(\Lambda)). \tag{11}$$

In parallel, the original singular values tensor $\sigma$ independently passes through additional convolutional layers and a Gaussian Error Linear Unit (GeLU) Hendrycks & Gimpel (2023) activation to generate context-sensitive modulation features $m_j$. It formulates as follows:

$$m_j = \text{GeLU}(\text{Convs}(\sigma)). \tag{12}$$

These context-sensitive characteristics $m_j$ are multiplied in element by the positional shrinkage ratios $r_j$, and the result is passed through a convolutional layer to produce the final adaptive shrinkage parameters $\gamma_j$, calculated as:

$$\gamma_j = \text{Conv}(r_j \cdot m_j). \tag{13}$$

The updated singular value is then computed as:

$$\sigma'_j = \text{ReLU}\left(\sigma_j - \gamma_j\right), \quad j = 1, 2, \ldots, K. \tag{14}$$

We reconstruct the low-rank component as $\hat{\mathcal{F}}_1^{(i)} = U\Sigma'V^H$, where $\Sigma' = \text{diag}(\sigma'_1, \ldots, \sigma'_K)$. The final solution to the $\mathcal{L}$ subproblem is given by:

$$\mathcal{L} = \Phi^H(\hat{\mathcal{F}}_1). \tag{15}$$

In this subsection, we introduce positional encoding to encode the relative position of each singular value and adapt the shrinkage behavior accordingly. PASS learns a position-dependent and content-aware shrinkage profile, enabling the model to differentiate between dominant anatomical features and redundant components based on their singular value position. To further support PASS design, we employ a multi-resolution training strategy where inputs are randomly cropped at different spatial scales. This encourages the model to generalize across different scales and reinforces the ability of PASS to learn shrinkage profiles that are sensitive to spatial context.

### 3.1.2 THE $\mathcal{S}$ LAYER WITH ATTENTION-BASED SOFT THRESHOLDING

For the $\mathcal{S}$ subproblem, we follow the design proposed in Zhang et al. (2025) and Zhao et al. (2023), which introduces an attention-based soft thresholding (AST) operator.

Given the transformed tensor $\mathcal{F}_2 = \Psi(\mathcal{X})$, AST first computes the channel-wise global average pooled vector $f \in \mathbb{R}^{N_c}$ from the absolute values $f = \text{GAP}(|\mathcal{F}_2|)$. This vector passes through two fully connected layers with ReLU and sigmoid activations to generate attention weights $\alpha_c \in [0,1]^{N_c}$. The thresholds $\tau_c$ are obtained by $\tau_c = \alpha_c \cdot f_c$. The shrinkage is applied as:

$$\hat{\mathcal{F}}_2^{[c]} = \text{Sign}(\mathcal{F}_2^{[c]}) \cdot \text{ReLU}(|\mathcal{F}_2^{[c]}| - \tau_c), \tag{16}$$

where $\mathcal{F}_2^{[c]}$ denotes the $c$-th channel of the data $\mathcal{F}_2$. The final sparse component is reconstructed by $\mathcal{S} = \Psi^H(\hat{\mathcal{F}}_2)$.

### 3.1.3 THE $\mathcal{X}$ LAYER WITH ADAPTIVE FUSION OF $\mathcal{L}\&\mathcal{S}$ COMPONENTS

The final step in each iteration updates the variable $\mathcal{X}$ by combining the low-rank and sparse estimates. Unlike conventional L+S models that apply a fixed or uniform fusion scheme, we adopt an $\mathcal{L}\&\mathcal{S}$ adaptive fusion (LSAF) strategy guided by local image features. This mechanism allows the network to dynamically adjust the influence of $\mathcal{L}$ and $\mathcal{S}$ at each spatiotemporal location, enhancing both structural consistency and detail preservation.

Specifically, we concatenate the low-rank $\mathcal{L}$ and sparse components $\mathcal{S}$ along the channel dimension. The concatenated tensor is passed through a learnable residual convolutional block $\text{Conv}_{\text{res}}$, followed by a sigmoid activation to produce spatially adaptive fusion weights:

$$\mathcal{W} = \text{Sigmoid}\left(\text{Conv}_{\text{res}}(\text{Concate}(\mathcal{L}, \mathcal{S}))\right). \tag{17}$$

These weights control the contribution of each component at every location. The updated variable $\mathcal{X}$ is computed as:

$$\mathcal{X} = \mathcal{W} \odot \mathcal{L} + (1 - \mathcal{W}) \odot \mathcal{S}. \tag{18}$$

Table 1: Reconstruction Performance on OCMR Dataset (single-coil). The metrics are reported by the mean PSNR/SSIM on the test set. The inference time is in seconds.

|  | radial-8 | radial-16 | radial-30 | vds-10 | vds-8 | TIME |
|---|---|---|---|---|---|---|
| PASSNet | **37.533/0.960** | **41.881/0.983** | **43.309/0.988** | **35.940/0.956** | **41.730/0.981** | 0.95 |
| JotlasNet | 36.505/0.952 | 41.708/0.982 | 43.231/0.987 | 34.979/0.949 | 41.671/0.981 | 0.87 |
| L+S-Net | 35.975/0.949 | 41.197/0.979 | 43.013/0.986 | - | - | 0.30 |
| ISTA-Net | 35.822/0.947 | 40.816/0.978 | 42.962/0.986 | 34.793/0.948 | 41.107/0.979 | 0.16 |
| DCCNN | 35.494/0.946 | 40.671/0.977 | 42.897/0.986 | 34.599/0.947 | 40.659/0.976 | **0.15** |
| SLR-Net | 34.173/0.937 | 38.635/0.964 | 40.212/0.971 | 33.880/0.934 | 39.056/0.968 | 0.34 |

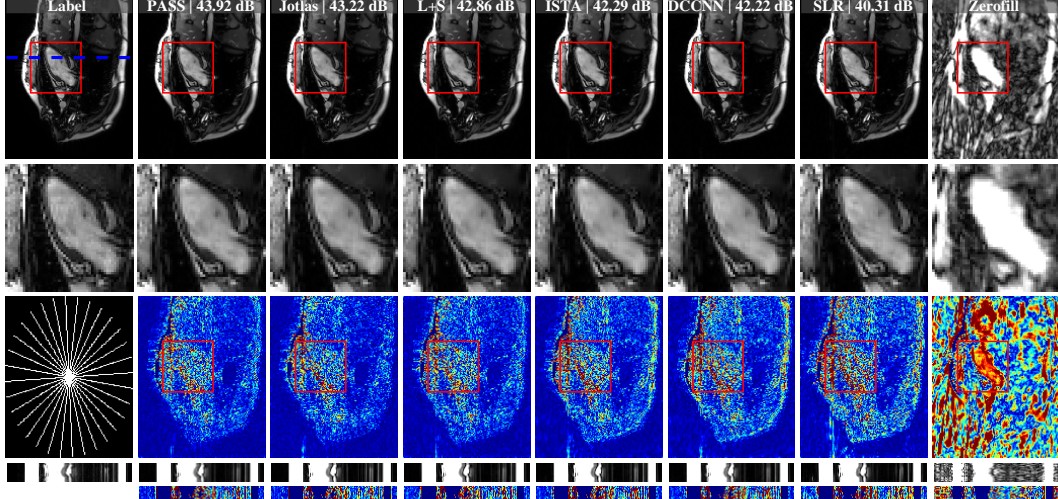

Figure 3: Reconstruction results under radial-16 sampling. Top to bottom: reconstructed frames, zoomed-in views, sampling mask and error maps, x-t images and x-t error maps. PSNR values are shown for each method.

This design allows the network to assign a higher weight to the low-rank term in regions with coherent structural patterns, such as organ interiors, and favor the sparse term in areas with localized variations or fine edges. The fusion module learns to balance these contributions in a data-driven manner, leveraging residual connections to stabilize training and ensure smooth integration between the two branches. By embedding this adaptive fusion step into the unfolded framework, our model achieves flexible regularization across space and time, improving reconstruction quality in both globally structured and locally detailed regions.

# 4 EXPERIMENTS AND RESULTS

## 4.1 COMPARISON EXPERIMENTS

We conduct extensive experiments on two publicly available cardiac MRI datasets, OCMR[1] Chen et al. (2020) and CMRxRecon[2] Wang et al. (2021a). For OCMR, we follow the data split and evaluate under single-coil settings with pseudo-radial and variable-density sampling masks. For CMRxRecon, we adopt the standard split and evaluate under multi-coil settings with 1D equispaced undersampling. More details of the training strategy and dataset preprocessing are provided in the Appendix. Our experiments includes a comprehensive comparative analysis against several state-of-the-art state-of-the-art dynamic MRI reconstruction DUN baselines. Peak Signal-to-Noise Ratio (PSNR) Huynh-Thu & Ghanbari (2008) and Structural Similarity Index (SSIM) Wang et al. (2004)

---

[1] https://www.ocmr.info/
[2] https://cmrxrecon.github.io/

Table 2: Reconstruction Performance on CMRxRecon Dataset (multi-coil). The metrics are reported by the mean PSNR/SSIM on the test set.

|            | 4X          | 8X          | 10X         |
|------------|-------------|-------------|-------------|
| PASSNet    | **37.947/0.951** | **35.689/0.946** | **34.942/0.943** |
| JotlasNet  | 37.796/0.946 | 35.298/0.939 | 34.631/0.937 |
| L+S-Net    | 37.525/0.940 | 35.034/0.931 | 34.348/0.929 |
| ISTA-Net   | 37.632/0.947 | 35.132/0.936 | 34.435/0.937 |
| DCCNN      | 37.106/0.931 | 34.358/0.913 | 33.898/0.910 |
| SLR-Net    | 37.098/0.934 | 34.302/0.920 | 33.808/0.915 |

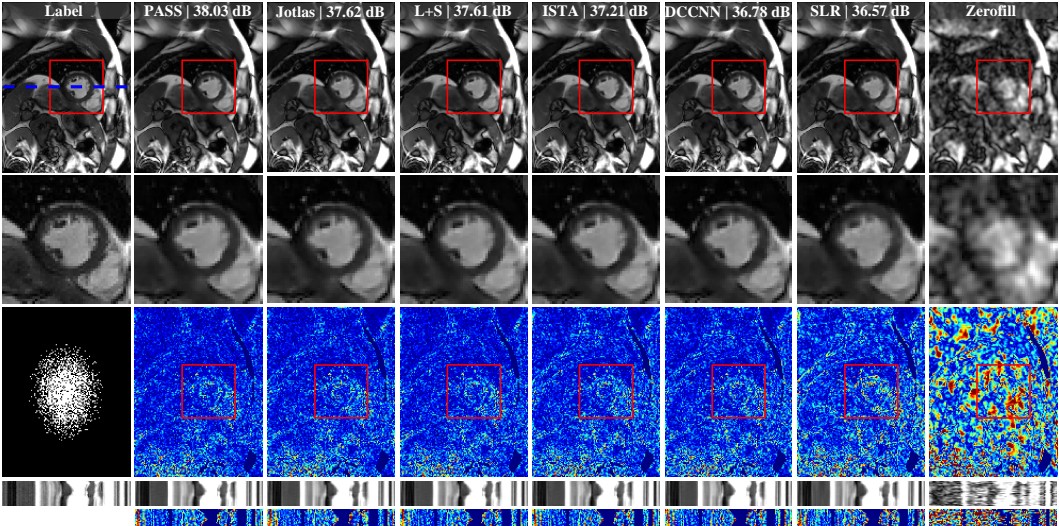

Figure 4: Reconstruction results under VDS-10 sampling. Top to bottom: reconstructed frames, zoomed-in views, sampling mask and error maps, x-t images and x-t error maps. PSNR values are shown for each method.

are used as the quantitative evaluation metrics across all experiments. The comparisons cover both OCMR and CMRxRecon datasets under various undersampling patterns and acceleration factors.

**Quantitative Results on OCMR**: Table 1 reports the reconstruction performance on the OCMR test set under radial and variable VDS sampling strategies. The proposed method PASSNet outperforms all baselines across all settings. In particular, it achieves a notable improvement of 1.028 dB in PSNR over JotlasNet under the radial-8 pattern, while maintaining competitive inference speed (0.95s per sample).

**Quantitative Results on CMRxRecon**: The performance comparison on the CMRxRecon dataset is shown in Table 2, using multi-coil data and 1D pseudoequispaced sampling masks. PASSNet again achieves the highest PSNR and SSIM values across all settings. At 10X sampling masks, our method achieves 34.942 dB / 0.943, outperforming JotlasNet by +0.311 dB and +0.006 in SSIM. These results demonstrate the scalability of our model to more challenging multi-coil, high-acceleration scenarios.

**Qualitative Visualization Comparison**: Figure 4 and Fig. 3 show representative reconstruction examples under radial-16 and vds-10 settings, respectively. Compared to baseline methods, PASSNet better preserves anatomical structures, especially at tissue boundaries and cardiac chambers. The residual error maps further highlight the advantage of our model. PASSNet produces the darkest error regions.

Table 3: Ablation study and iteration stage analysis. Left: effectiveness of PASS, LSAF, and multi-resolution strategy. Right: PSNR/SSIM/Parameter under different iteration numbers in radial-16.

| Config | PASS | LSAF | Multi-R | radial-8 | vds-10 | Iter num | PSNR | SSIM | Param |
|--------|------|------|---------|----------|--------|----------|------|------|-------|
| I | ✓ | × | × | 36.61/0.95 | 35.02/0.94 | 9 | 41.129 | 0.980 | 0.75M |
| II | × | ✓ | × | 37.15/0.95 | 35.64/0.95 | 13 | 41.577 | 0.982 | 1.09M |
| III | ✓ | ✓ | × | 37.21/0.95 | 35.72/0.95 | 15 | 41.881 | **0.983** | 1.25M |
| IV | ✓ | ✓ | ✓ | **37.53/0.96** | **35.94/0.96** | 17 | **41.919** | 0.982 | 1.42M |

## 4.2 ABLATION STUDY

We conduct comprehensive ablation study on the OCMR test set to analyze the contribution of three key design. Quantitative results are reported in Table 3.

**Effectiveness of PASS and LSAF Modules**: To evaluate the individual contributions of the proposed PASS and LSAF modules, we replace them with baseline alternatives: the soft-thresholding function and direct addition of $\mathcal{L}$ and $\mathcal{S}$ components, respectively. In Table 3, configurations I, II represent models with specific components removed. The full model (Ours-IV) integrates both modules.

Results show that removing either module degrades performance. Under the vds-10 sampling pattern, excluding PASS (Config I) results in a 0.920 dB drop in PSNR (2.6%) and a 0.008 reduction in SSIM (0.8%). Excluding LSAF (Config II) leads to a PSNR drop of 0.298 dB (0.8%) and SSIM reduction of 0.003 (0.3%). These findings confirm the individual and combined effectiveness of both modules, highlighting their complementary roles in the framework.

**Multi-Resolution Training Strategy** To evaluate the effectiveness of the proposed multi-resolution training strategy, we compare it with a fixed-resolution baseline. In the baseline setting, we follow the same dataset split and data augmentation protocol as in Zhang et al. (2025), where all training samples are uniformly cropped to 128×128×16. As shown in Table 3, the model trained with fixed resolution (Config III) underperforms the full model (Ours-IV) across both radial-8 and vds-10 sampling patterns. This suggests that multi-resolution training improves reconstruction quality. Multi-resolution training strategy complements the PASS module, which incorporates position encoding into the adaptive shrinkage process. By training the model from varied spatial regions and resolutions, the network becomes more sensitive to positional differences.

**Effect of Unfolded Stage Number**: To investigate the impact of the number of iterative stages, we evaluate the model under the radial-16 sampling pattern with different numbers of unfolded stage. As shown in Table 3, performance improves steadily as the number of stages increases from 9 to 13. Further increasing the stage count to 17 results in marginal gains with fluctuations in PSNR and SSIM, indicating diminishing returns. Additionally, more stages lead to increased parameter count and computational cost during both training and inference. Therefore, we choose 15 stages as the default setting to balance performance and efficiency.

## 5 CONCLUSION

This paper proposes a PASS mechanism for dMRI reconstruction, which enhances SVT with spectral positional encoding and learnable gating to achieve adaptive singular value shrinkage. Integrated into an unfolded L+S framework with an LSAF module, the method dynamically balances low-rank and sparse components based on local features. Extensive experiments on multiple datasets show that PASSNet consistently outperforms existing DUN baselines under various sampling patterns and acceleration factors, demonstrating superior accuracy, robustness, and generalization. In future work, we plan to explore the application of our method to other domains such as hyperspectral imaging.

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

# A ADDITIONAL DETAILS

## A.1 TRAINING STRATEGY

The proposed PASS-Net is trained in a supervised learning framework using the pixelwise mean squared error (MSE) as the loss function:

$$\mathcal{L} = \frac{1}{M} \sum_{n=1}^{M} \|\hat{\mathcal{X}} - f_{\text{net}}(\mathscr{A}, \mathbf{d}|\theta)\|_F^2, \quad (19)$$

where $\hat{\mathcal{X}}$ is the fully sampled ground truth, $\theta$ denotes the trainable parameters of the network, $M$ is the number of training samples, and $f_{\text{net}}(\mathscr{A}, \mathbf{d}|\theta)$ represents the network's reconstruction output given the undersampling operator $\mathscr{A}$ and k-space data $\mathbf{d}$.

The model is implemented in TensorFlow and optimized using Adam Kingma & Ba (2017) with $\beta_1 = 0.9$, $\beta_2 = 0.999$, and $\epsilon = 10^{-8}$. The learning rate starts at $1 \times 10^{-3}$ and decays by 0.95 per epoch. Training runs for 50 epochs with a batch size of 1. The network includes 15 unrolled stages and 1.25M trainable parameters, with all convolutional and fully connected layers using 16 channels. During training, a pseudo-random sampling mask is generated at each step to simulate various sampling types and acceleration factors. In addition, we adopt a multi-resolution strategy where each epoch randomly presents inputs of different spatial sizes. This variation helps the network learn across diverse anatomical scales and improves the generalization of the position-aware PASS module. Details of the resolution configurations and data preparation process are provided in next section. All experiments are conducted on a workstation with an Intel(R) Xeon(R) Platinum 8368 CPU and NVIDIA A100-SXM4 GPU (40GB).

## A.2 DATASET AND BENCHMARK

OCMR dataset consists of 204 fully sampled cardiac cine MRI raw data acquired from 74 subjects using three Siemens MAGNETOM scanners (3T Prisma, 1.5T Avanto, and 1.5T Sola), including both short-axis and long-axis views. We follow the data split provided by JotlasNet Zhang et al. (2025), with 124/40/40 cases for training, validation, and testing. To enhance spatial diversity and support the position-aware design of the PASS module, we crop training volumes into spatial resolutions of [64×64×16, 96×96×16, 128×128×16, 144×144×16, 156×156×16] with strides of [32,32,8]. This processes results in a total of 1,764 training samples. We conduct experiments under the single-coil scenario using two sampling: pseudo-radial sampling Lingala et al. (2011) with 8, 16, and 30 lines, and variable density random sampling (VDS) with acceleration factors of 8 and 10.

CMRxRecon dataset contains 120 fully sampled multi-contrast dynamic cardiac images with raw k-space data collected from a 3T scanner, including both LAX and SAX views. We adopt a 10:1:1 split, yielding 1,247, 124, and 121 fully sampled multi-coil samples for training, validation, and testing, respectively. We do not apply cropping, and the raw spatial dimensions vary across samples. This naturally introduces multi-resolution characteristics into training. Experiments are conducted under the multi-coil scenario using 1D pseudo-equispaced sampling Zbontar et al. (2019); Knoll et al. (2020) with acceleration factors of 4×, 8×, and 10×.

The single-coil inputs and ground-truth labels are both derived using ESPIRiT Muckley et al. (2021) coil sensitivity estimation. We evaluate the proposed method against several state-of-the-art DUNs for dynamic MRI reconstruction, including SLR-Net Ke et al. (2021), DCCNN Schlemper et al. (2018), ISTA-net Zhang & Ghanem (2018), L+S-net Huang et al. (2021), and JotlasNet Zhang et al. (2025).

