# OpenReview forum: "Position-Aware Singular Value Shrinkage for Unfolded Dynamic MRI Reconstruction"
_ICLR.cc/2026/Conference — Submitted to ICLR 2026_

### Official Review · Reviewer_UzLA · 2025-10-31

**Soundness:** 3
**Presentation:** 3
**Contribution:** 3
**Rating:** 4
**Confidence:** 3

**Summary:**

The paper proposes a Position-Aware Adaptive Singular-Value Shrinkage (PASS) module and embeds it into an L+S-unfolding framework to enhance the discriminative ability of singular-value thresholding.

**Strengths:**

1.Clear motivation. The paper clearly identifies the limitation of uniform thresholds and the resolution-dependent variation of singular spectra, addressing a real bottleneck in dMRI reconstruction.
2.Well-structured design. The modular layout of PASS, AST, and LSAF is easy to follow and practical for implementation or replacement.

**Weaknesses:**

1.Weak theoretical support. The paper lacks a solid explanation of why positional encoding helps and under what conditions it is beneficial; most evidence is empirical.
2.Unclear implementation details. The interaction between positional encoding and convolution in PASS is not clearly described — the handling of the K dimension, convolution type, and threshold generation remain vague.
3.Limited validation. Cross-resolution or sampling-rate generalization experiments are missing, making it hard to assess true adaptivity.

**Questions:**

1.The mathematical definition of the position encoding in PASS is ambiguous and lacks correspondence to the standard sinusoidal positional encoding. The paper should clarify its formulation and explicitly explain how the proposed encoding relates to, or differs from, classical positional encodings.
2.Although the paper claims to learn position- and magnitude-aware thresholds for singular values, the mechanism essentially performs a learnable re-weighting of the singular spectrum. The claimed "position-aware" nature is not clearly demonstrated either theoretically or empirically.
3.The paper does not provide a mechanistic or theoretical explanation for the effect of positional encoding within PASS. Most of the presented evidence relies on downstream loss improvements. It is strongly recommended to include explicit mechanism experiments or ablation studies isolating the effect of positional encoding.
4.The tensor Λ is described as having a shape of B×T×K×2, but the paper does not specify whether convolution is performed along the K dimension (1D convolution) or if K is treated as a channel dimension. Since K varies with spatial resolution, this directly affects parameter invariance and reproducibility. Details such as convolution dimension, kernel size, and padding should be clearly stated, along with how the method adapts to variable K.
5.Table 3 indicates that the number of parameters changes with the unfolding stage count, while Appendix A.1 claims the total parameter size is fixed at approximately 1.25M (for 15 stages). This inconsistency needs to be resolved, and the actual parameter dependence on stage count should be clarified.
6.The experimental section lacks details on hardware configuration, batch size, and implementation settings. Moreover, the use of OCMR data is confusing: the paper refers to a "single-coil" scenario but also mentions that both the input and ground truth are estimated via ESPIRiT coil sensitivity maps, which contradicts the single-coil assumption. The authors should clarify the exact data preprocessing and reconstruction setup.

---

### Official Review · Reviewer_ztmG · 2025-10-31

**Soundness:** 3
**Presentation:** 3
**Contribution:** 3
**Rating:** 6
**Confidence:** 4

**Summary:**

This paper presents a Position-Aware Adaptive Singular-Value Shrinkage (PASS) mechanism for dynamic MRI (dMRI) reconstruction within a deep unfolding framework. PASS enhances conventional Singular Value Thresholding (SVT) by introducing spectral positional encoding and a learnable gating mechanism, enabling context-sensitive and position-dependent shrinkage of singular values. The authors integrate PASS into an L+S decomposition-based unfolding network, together with an Adaptive Fusion (LSAF) module and a multi-resolution training strategy to improve adaptability across varying anatomical scales and acquisition settings. Extensive experiments on two cardiac MRI datasets demonstrate clear improvements over strong baselines in terms of PSNR and SSIM.

**Strengths:**

- The introduction of a position-aware adaptive shrinkage mechanism for singular values is both novel and well-motivated. It directly addresses the limitation of uniform SVT in low-rank priors for dynamic MRI.
- The integration of PASS with L+S decomposition, adaptive regularization, and LSAF module forms a synergistic system. The multi-resolution training strategy further enhances the model’s adaptivity to varying spatial scales and acquisition settings, addressing a critical gap in existing methods.
- The method focuses on dynamic cardiac MRI reconstruction, a clinically critical task, and demonstrates superior preservation of fine anatomical structures.
- The paper is generally well-written. The problem statement is clear.

**Weaknesses:**

- All experiments focus on cardiac MRI. The claim of general applicability lacks supporting experiments or discussion about transferability.
- The method is only validated on cardiac MRI datasets. While the results are strong, it remains to be seen how well it generalizes to dynamic MRI of other anatomies (e.g., brain, abdomen) with different spatiotemporal characteristics.
- The comparisons are limited to other Deep Unfolding Networks (DUNs). It would be more compelling to also compare against recent non-unrolled deep learning models (e.g., transformer-based or other pure CNN architectures)
- The sinusoidal positional encoding used in PASS is adopted without justification. The authors do not compare it with alternative positional representations (e.g., absolute encoding, learnable positional encoding) or provide ablation studies on key parameters (e.g., the scaling factor 10000 in the encoding formula).

**Questions:**

- How does the inclusion of PASS affect the overall computational cost per iteration? Was a truncated or approximate SVD used to ensure tractable inference time?
- Why was a sinusoidal encoding function (Eq. 9) chosen over a learnable or linear encoding? How does it affect stability during multi-resolution training?
- Can the authors visualize or analyze how PASS modifies singular value distributions? For instance, how do the shrinkage ratios vary with singular value rank?
- How does the model perform under different sampling schemes (e.g., Cartesian, spiral) or varying noise levels? Does the learned adaptivity remain effective?

---

### Official Review · Reviewer_8Xrc · 2025-10-31

**Soundness:** 3
**Presentation:** 2
**Contribution:** 1
**Rating:** 2
**Confidence:** 5

**Summary:**

This paper introduces Position-Aware Adaptive Singular-Value Shrinkage (PASS) for dynamic MRI reconstruction. PASS replaces the conventional uniform Singular Value Thresholding (SVT) within low-rank regularized deep unfolding networks by incorporating spectral positional encoding and a neural gating mechanism. The authors embed PASS in a low-rank + sparse (L+S) unfolded network, add an adaptive fusion (LSAF) module, and use a multi-resolution training strategy. Experiments on OCMR and CMRxRecon datasets show improved PSNR/SSIM over baselines (e.g., JotlasNet, L+S-Net, ISTA-Net).

**Strengths:**

1. The paper addresses the well-known limitation of global/uniform SVT thresholds that ignore spectral structure and resolution variance.

2. The introduction of positional encoding within singular-value space is conceptually fresh and technically consistent.

3. The paper includes experimental results with two public datasets, multiple undersampling patterns, and ablations (PASS, LSAF, multi-resolution).

4. Training/inference details, datasets, and hardware are clearly documented, which is commendable for reproducibility.

**Weaknesses:**

1. Marginal technical novelty relative to JotlasNet and L+S-Net.
- The main idea (adaptive SVT + L+S unfolding) is essentially an incremental refinement over [Zhang et al., MRI, 2025] and [Huang et al., MedIA, 2021].
- The "position-aware" aspect is achieved by concatenating sinusoidal embeddings to singular values---this is minor and not theoretically grounded.
- The gating network in Eqs. (11)-–(13) is a lightweight convolutional module with no clear justification for how it captures spatial or anatomical context beyond empirical tuning.

2. Weak theoretical or interpretive support
- No analytical evidence or ablation on why spectral position correlates with anatomical relevance.
- The sinusoidal encoding in Eq. (9) seems arbitrary (mirroring transformer-style token positions) without medical or mathematical rationale.

3. Experimental margin too small for ICLR standard
- Gains over JotlasNet are modest: +0.3--1.0 dB PSNR and +0.003--0.008 SSIM.
- The ablation study (Table 3) shows that multi-resolution training contributes as much as PASS itself, suggesting improvements stem from training setup rather than the proposed mechanism.
- Inference time is actually slower (0.95 s vs 0.87 s).

4. No comparison with non–unfolding or transformer-based MRI methods
- Missing recent high-performing baselines such as diffusion priors or transformer-based reconstruction (e.g., VarNet++ and SwinMR).
- As ICLR increasingly favors conceptually generalizable models, the work feels limited to one niche (dMRI unfolding).

5. Writing and figure clarity issues
- Fig. 2 is crowded and unreadable; module labels (a)–(c) are insufficiently explained.
- Minor grammatical issues persist throughout, likely due to automated translation.

**Questions:**

1. On the necessity of positional encoding in the singular-value domain:
The paper introduces sinusoidal positional embeddings in Eq. (9) to represent the "relative importance" of each singular value.
- How was this specific encoding (sin(linspace / 10000^{j/K})) chosen?
- Did you test simpler formulations (e.g., linear rank normalization or learned embeddings)?
- Can you provide evidence that the positional information---not just the gating MLP---drives the observed improvement?

2. On the interpretability of PASS:
The paper claims that large singular values correspond to dominant anatomy while small ones represent noise or subtle textures.
- Did you visualize or quantify this claim (e.g., by plotting reconstructed components corresponding to different $\sigma$ ranges)?
- How robust is the learned thresholding when anatomical motion or contrast distribution changes across subjects?

3. On fairness of baseline comparison:
The baselines (ISTA-Net, DCCNN, L+S-Net, and JotlasNet) are all unfolding-based.
- Why are transformer-based or diffusion-prior reconstruction methods (e.g., VarNet++, SwinMR, or recent fastMRI Challenge winners) excluded?
- Without such comparisons, how can we evaluate PASSNet's competitiveness at ICLR scale?

4. On ablation and contribution isolation:
Table 3 shows that multi-resolution training contributes similar or larger gains than the PASS module itself.
- Could you provide a quantitative breakdown showing which component contributes most across datasets?
- Is PASS still beneficial when multi-resolution training is disabled?

5. On computational efficiency:
PASSNet has ~1.25 M parameters but ~5–6x slower inference (0.95 s vs. 0.16 s for ISTA-Net).
- Have you profiled the runtime bottleneck?
- Would using a low-rank approximation of the gating network mitigate the cost without performance loss?

6. On generalization beyond cardiac MRI:
The paper mentions potential extension to hyperspectral imaging.
- Have you tested PASSNet on any non-cardiac sequences (e.g., brain dMRI, flow MRI)?
- Does the learned positional pattern transfer, or must PASS be retrained per anatomy?

---

### Official Review · Reviewer_Gxox · 2025-11-01

**Soundness:** 2
**Presentation:** 2
**Contribution:** 2
**Rating:** 4
**Confidence:** 3

**Summary:**

The paper targets dynamic MRI reconstruction within deep unfolding (DUN) frameworks that impose low-rank priors via Singular Value Thresholding (SVT). It argues that the widespread use of uniform/global thresholds overlooks the unequal importance of singular values and resolution dependence, leading to either over-smoothing details or insufficient noise removal. To address this, the authors propose PASS (Position-Aware Adaptive Singular-value Shrinkage): a learnable module that augments SVT with spectral positional encoding and a neural gating mechanism to produce per-value adaptive shrinkage (γ_j). PASS is integrated into an L+S (low-rank + sparse) unfolded network, with an L&S adaptive fusion (LSAF) step that balances L and S based on local image features; a multi-resolution training strategy further improves robustness across scales.Experiments on OCMR (single-coil) and CMRxRecon (multi-coil) demonstrate sustained improvements in PSNR/SSIM compared to DUN baselines (e.g., JotlasNet, L+S-Net, ISTA-Net, DCCNN, SLR-Net). Ablation experiments validate the contributions of PASS, LSAF, and multi-resolution training. However, among the baselines compared, only the JotlasNet paper is from 2025; the rest are from before 2021 and cannot be considered SOTA.

**Strengths:**

Originality: PASS replaces one-size-fits-all SVT with learned, per-value adaptive shrinkage informed by positional encodings; LSAF provides spatially adaptive L-S fusion—together a clean, effective upgrade to L+S unfolding.
Quality: Strong experimental breadth (two public cardiac dMRI datasets; multiple patterns/factors), clear wins in PSNR/SSIM, and ablation + stage analysis that isolate each component’s contribution.
Clarity: Method equations and block diagrams (PASS/AST/LSAF) are explicit; the reconstruction operator is defined; training details are documented.
Significance: Addresses real clinical needs (structure fidelity, cross-resolution robustness) with consistent gains under both single- and multi-coil settings.

**Weaknesses:**

a.Statistics & metric diversity are limited. Results rely mainly on mean PSNR/SSIM; variance, confidence intervals, and statistical significance are not reported; NRMSE/HFEN/k-space errors are absent.
	b.Comparisons to classical adaptive shrinkage are mostly conceptual in text; direct experimental baselines (e.g., ATN, NNFN) under the same settings are missing.
	c.OOD robustness not systematically explored. The paper motivates resolution/anatomy dependence but lacks controlled shift studies (e.g., altered pixel spacing/FOV, anatomical variance, noise mismatch).
	d.Complexity transparency. Inference times are mentioned, but module-level FLOPs/latency/memory and the effect of removing PASS are not broken down.

**Questions:**

a.The SOTA claim is not credible given the baseline vintage. With the exception of a single 2025 paper, your comparisons rely predominantly on methods from 2021 or earlier. This is not acceptable for a 2025 submission in dynamic/cardiac MRI reconstruction. The field has advanced materially since 2022 (unfolding upgrades, learned DC, plug-and-play priors, transformer/SSM, self-supervised regimes). Absent head-to-head comparisons against contemporaneous methods (2022–2025), the manuscript cannot substantiate a state-of-the-art claim. At best, it shows superiority over outdated baselines.
	b.What inductive bias does the positional encoding capture (e.g., relative rank vs. magnitude)?
	c.Any empirical evidence that PASS does not destabilize the L/S updates across stages (e.g., monotonic residual decay, Lipschitz bounds, or contraction proxies)?
	d.Provide training curves of loss / validation PSNR vs. iterations to show stability.
	e.How sensitive is performance to (i) the number of unfolding stages, (ii) PASS strength (e.g., gating temperature/scale), and (iii) the positional encoding dimensionality?
	f.Why can’t a well-tuned classical adaptive rule match PASS if embedded in the same unfolding pipeline?
	g.Please add a controlled baseline that replaces PASS with a classical adaptive shrinkage (ATN/NNFN or a differentiable proxy) within your unfolding.
	h.You ablate PASS, LSAF, and #stages. Could you also show “No multi-resolution training” vs “With” to quantify its standalone effect?
	i.Which SVD is used (full/truncated/batched)? On what tensor shapes?Would a low-rank approximation or randomized SVD materially change accuracy/latency?

---

### Meta-Review · Area_Chair_advi · 2026-01-08

**Summary:**

The paper introduces the Position‑Aware Adaptive Singular‑Value Shrinkage (PASS) module for dynamic MRI reconstruction within deep unfolding networks. PASS enhances conventional Singular Value Thresholding (SVT) by incorporating spectral positional encoding and a neural gating mechanism, enabling adaptive shrinkage of singular values. Integrated into a low‑rank plus sparse (L+S) unfolding framework with an adaptive fusion module and multi‑resolution training, the method shows improvements in PSNR and SSIM on cardiac MRI datasets compared to existing SVT‑based baselines.

**Reviewer Concerns:**

The reviewers have concerns that the contribution is incremental, as adaptive SVT and L+S unfolding have been explored in prior work. Methodology was questioned, with insufficient theoretical justification for positional encoding, vague implementation details, and missing ablations on its effect. Comparisons were limited to older unfolding‑based baselines, excluding more recent transformer or diffusion‑based MRI methods. Results are improved modestly, with generalization only tested on cardiac MRI. Reviewers also noted missing evaluations on other anatomies, newer datasets, and computational efficiency analysis.

**Reviewer Scores:**

The submission received mixed ratings: 4, 2, 6, and 4. Two reviewers (scores 2 and 4) judged the work marginally below acceptance due to limited novelty, weak theoretical support, and outdated baselines. One reviewer (score 6) found the method marginally above the threshold, appreciating the introduction of position-aware adaptive shrinkage, clinical relevance and strong cardiac MRI results. The authors did not provide responses to these concerns, therefore cannot be accepted.

---

### Decision · Program_Chairs · 2026-01-26

Reject